# The Oxidation Behaviors of Indefinite Chill Roll and High Speed Steel Materials

**Liang Hao, Tuanjie Li \*, Zhongliang Xie, Qingjuan Duan and Guoyuan Zhang**

School of Mechano-electronic Engineering, Xidian University, Xi'an 710071, China; haoliang@xidian.edu.cn (L.H.); zlxie@xidian.edu.cn (Z.X.); qjduan@126.com (Q.D.); gyzhang@xidian.edu.cn (G.Z.)

**\*** Correspondence: tjli@mail.xidian.edu.cn; Tel.: +86-931-2976-688; Fax: +86-931-2976-578

**Abstract:** Indefinite chill (IC) roll and high speed steel (HSS) materials have been widely employed to manufacture work rolls as latter and former stands in hot rolling mills. The oxidation of work rolls is of importance for the surface quality of the rolled workpieces. The isothermal oxidation of the IC and HSS materials was conducted at 650 °C and 700 °C in both dry air and humid air. The isothermal oxidation curves indicate that HSS shows faster kinetics than the IC materials in dry air, whereas the opposite occurred in humid air. The oxide scales of the IC materials after the oxidation in both dry air and humid air are made up of two oxide phases. Two oxide phases were found when the HSS oxidized in the dry air and three oxide phases were found when oxidized in the humid air.

**Keywords:** indefinite chill roll; high speed steel; oxidation behavior; thermogravimetric analyzer (TGA)

## 1. Introduction

Hot rolling is one of the primary manufacturing processes in the steel industry, which is highly dependent on the property of work rolls. In the beginning, low-carbon and adamite rolls were applied as standard work rolls. However, the absence of carbides in low-carbon rolls led to a low wear resistance and a frequent replacement of rolls. As for the adamite rolls, a rapid heat crack formation and propagation takes place and requires an immediate roll change and redressing [1–3]. The introduction of high chromium (Hi-Cr) steel, high speed steel (HSS) and indefinite chill (IC) materials has made tremendous improvements in roll performance and campaign length. Hi-Cr rolls with 18% Cr are oxidation-resistant and result in the attachment of rolled materials onto work rolls. Adjustments were made to reduce Cr to less than 7% and add Mo, V, W and C to develop HSS [4–7]. They present high wear resistance and strength but are still difficult to oxidize. Further modifications were made to develop IC roll materials, including a considerable amount of graphite, showing a good thermal and oxidation behavior [8–10]. In addition, the graphite in the IC rolls can retard crack propagation and overcome sticking, since it lubricates the contact between the roll and the rolled strips [11–13]. Nevertheless, the hardness of IC rolls is less than that of HSS, so the wear loss for IC rolls is more rapid. Global experience recommends using HSS rolls primarily in former stands, such as F1–F3, and to use IC rolls in later stands in the hot strip mills [14]. Because of the multiphased microstructures in the HSS rolls, there is no clear consensus regarding the oxidation behavior of each phase at the rolling temperatures [15–17]. For instance, Kim et al. [18] proposed that carbides as well as the martensitic matrix be oxidized in a dry atmosphere, and only use a matrix oxidized in a wet atmosphere. Molinari et al. [19] suggested that the oxidation nucleates at the matrix–carbide interfaces and propagates in the matrix without involving the carbides, because of the high oxidation resistance of carbides. Later, they [5] added that some of the carbides do oxidize but show a different oxidation behavior

from the matrix. Particularly, $M_7C_3$ (Cr-rich) carbides do not oxidize at the rolling temperatures, MC (V-rich) carbides display a faster oxidation and $M_2C$ (Mo-rich) carbides show an intermediate behavior. Zhou et al. [20] found that only the matrix and the MC carbides oxidized but the $M_7C_3$ carbides hardly oxidized, and the effect of water vapor temperature on the morphology is negligible at each oxidation temperature [21]. Garza et al. [22] thought that $M_6C$ (Mo-rich) and $M_7C_3$ do not oxidize in humid atmospheres but MC carbides show a significant oxidation in a dry atmosphere.

During the hot rolling, the heat conduction from hot strips, deformation heat, and friction heat can result in a 700 °C surface temperature of work rolls in a very short contact time ($10^{-2}$–$10^{-3}$), which can then be cooled to room temperature by spraying water in the rest rime of each cycle ($2$–$10^{-1}$) [23]. The accumulated time of work rolls in high temperature is less than 30 min in periods of roll replacement. Furthermore, the oxide scales formed on the roll surface peels off due to friction and thermal fatigue [24–29]. The purpose of this paper is to investigate and compare the oxidation behavior of the IC and the HSS roll materials by a combined study of the oxidation kinetics, surface morphology and cross sections of the oxide scales formed in both dry and humid atmospheres.

## 2. Experimental

The IC roll and HSS materials were cut from the shell part of work rolls, and their chemical compositions are listed in Table 1. Samples were machined to the dimensions of $15 \times 10 \times 1$ mm$^3$ with a 0.5 mm side length square hole near the top edge of the sample. Two broad surfaces were polished up to a 1 um diamond suspension and the rest surfaces were ground with 1200 grit sand paper. Then, the samples were ultrasonically cleaned in acetone and rinsed with alcohol.

**Table 1.** Chemical compositions of the indefinite chill (IC) and high speed steel (HSS) materials (wt.%).

| Material | Fe | C | Ni | Cr | V | Mo | Si | Mn |
|----------|---------|------|------|------|-----|------|------|------|
| IC | Balance | 3.35 | 4.54 | 1.85 | - | 0.49 | 0.77 | 0.89 |
| HSS | Balance | 2.1 | - | 4.5 | 4.4 | 4.9 | 0.62 | 0.55 |

The isothermal oxidation was carried out on a SETSYS Evolution S60/58507(SETARAM Inc., Marseille, France), which is equipped with a vertical hang-down SETSYS balance. The oxidation tests were conducted under two atmospheres, 20% water vapor and dry air. The isothermal oxidation was chosen at 650 and 700 °C for 30 min, respectively [30]. The experimental details are given as follows: (a) the sample was suspended in the furnace chamber, and then the furnace chamber was vacuumed and flushed with Ar until the pressure reached 1 atm; (b) the sample was heated to the testing temperatures at a rate of 25 °C/min and held for 5 min once the testing temperature was reached; (c) the oxidizing gas was introduced and the isothermal oxidation processed for 30min; (d) the isothermal oxidation was terminated by replacing the oxidation gas with Ar to prevent further oxidation and the sample was cooled to room temperature at a rate of 30 °C/min. The mass variations were recorded in this process. The air containing 20% water vapor was obtained by setting the temperature of the water tank at 60 °C [31].

The microstructure analysis was employed on a JSM scanning electron microscope (SEM) equipped with an energy dispersive spectrometer (EDS) (JEOL Ltd., Tokyo, Japan). The morphology and phases of the oxide scales were studied both on the exposure surfaces and the cross sections. In addition, the oxidation kinetics curves were obtained by plotting the mass gain change per surface against the oxidation time.

## 3. Results

### 3.1. Microstructure of IC Roll and HSS Materials

Figure 1 shows the backscattered electron (BSE) images of the polished IC and HSS materials prior to the tests, respectively. Three different phases can be identified in the IC material (Figure 1a),

and they are graphite, (Fe, Cr)$_3$C (cementite), and martensite, respectively [32]. The HSS (Figure 1b) presents different microstructure characteristics. Three different carbides can be clearly distinguished by their contrasts, morphologies and EDS analysis. The dark areas are V-rich MC carbides, the grey regions are Cr-Fe-rich M$_7$C$_3$ carbides, and the white long zones are Mo-rich M$_6$C carbides [7,22,33]. These microstructure characteristics enable the IC and HSS materials to present special performances.

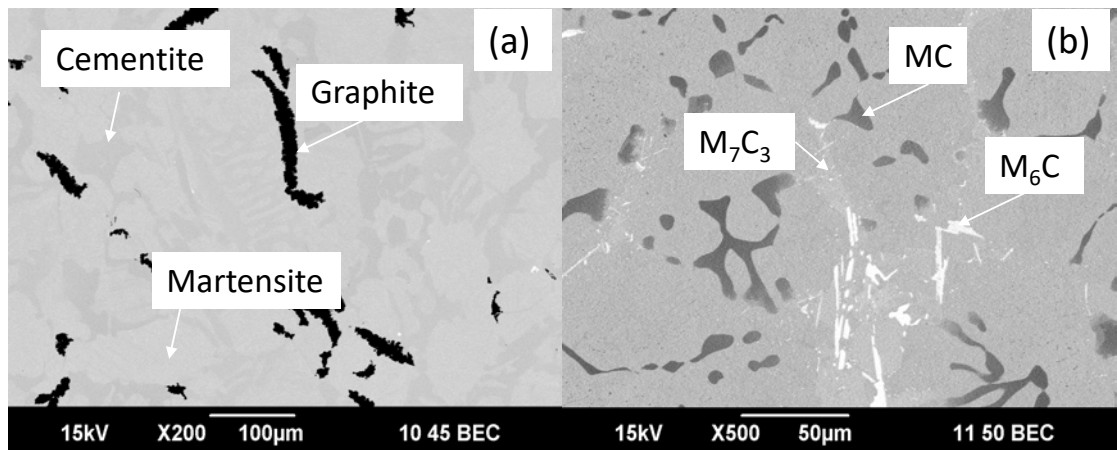

**Figure 1.** (**a**) Backscattered electron (BSE) images of the polished indefinite chill (IC) and (**b**) high speed steel (HSS) materials prior to the tests.

*3.2. Oxidation Kinetics*

The isothermal oxidation curves of the IC and HSS materials in both dry and humid air are shown in Figure 2. It is evident that both the temperature and oxidizing atmospheres obviously influence the mass gain of the materials. In dry air (Figure 2a), the mass gain of the HSS is greater than that of the IC rolls at corresponding temperatures, and reveals a parabolic law, while the mass gain of the IC rolls approximately demonstrates a linear law. In addition, the presence of water vapor (Figure 2b) accelerates the mass gain of the IC rolls more than that of the HSS, whereas the oxidation kinetics are little changed by the water vapor.

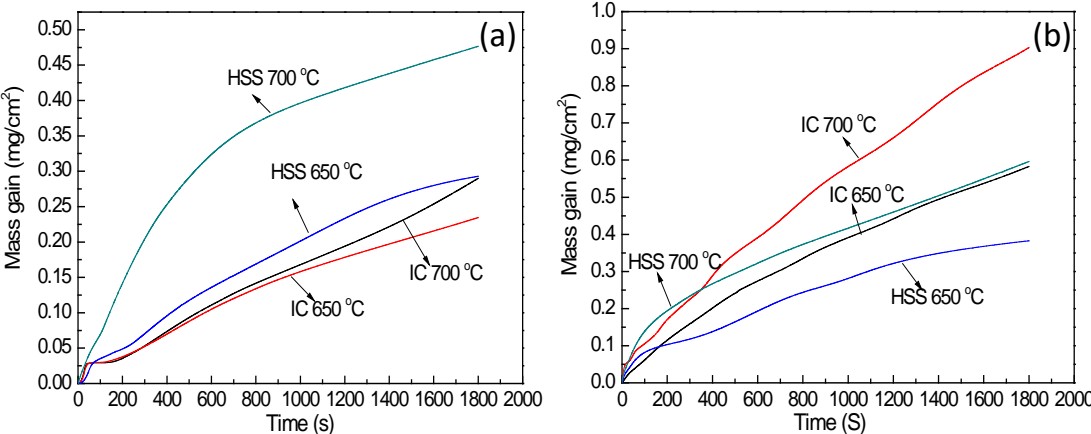

**Figure 2.** Isothermal oxidation curves of the IC and HSS materials in both (**a**) dry and (**b**) humid air.

*3.3. Surface Morphologies*

3.3.1. Oxidized in the Dry Air

Figure 3 shows the SEM images of the IC roll surface morphologies oxidized in the dry air at 650 °C and 700 °C. At 650 °C (Figure 3a), the martensite seems to oxidize faster than that of the cementite,

and the oxide scales on the martensite protruded out, since the cementite contains higher Cr content and is more oxidation-resistant than the martensite. The extension of the oxide scales nearly covers the graphite areas. When the oxidation processed at 700 °C (Figure 3b), the oxide scales approximately covered the entire surface and obscured the initial boundaries between the cementite and martensite. The cracks were observed in some zones (circled area).

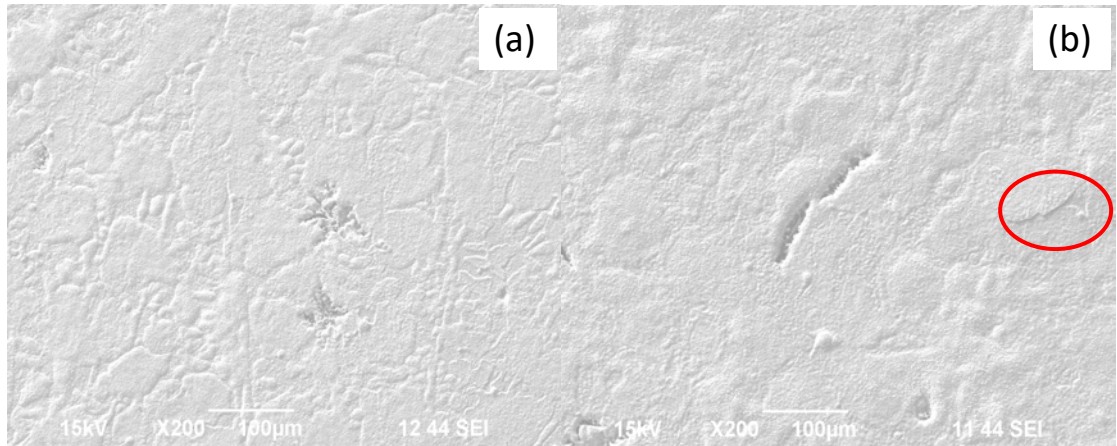

**Figure 3.** SEM images of the IC roll surface morphologies oxidized in dry air at (**a**) 650 °C and (**b**) 700 °C.

Similar phenomena were observed for the HSS oxidized at 650 °C and 700 °C in the dry air. SEM X-ray maps of the HSS oxidized at 700 °C in the dry air are presented in Figure 4, from which it can be seen that the areas of low O Ka intensity correspond to the areas of the carbides while the areas with a high O Ka intensity are of the matrix. The oxidation of MC carbides extends laterally, whereas the $M_7C_3$ carbides are still not totally covered by the oxides. Therefore, the matrix oxidized more readily than that of the carbides, and the $M_7C_3$ carbides are more resistant to oxidization than MC carbides.

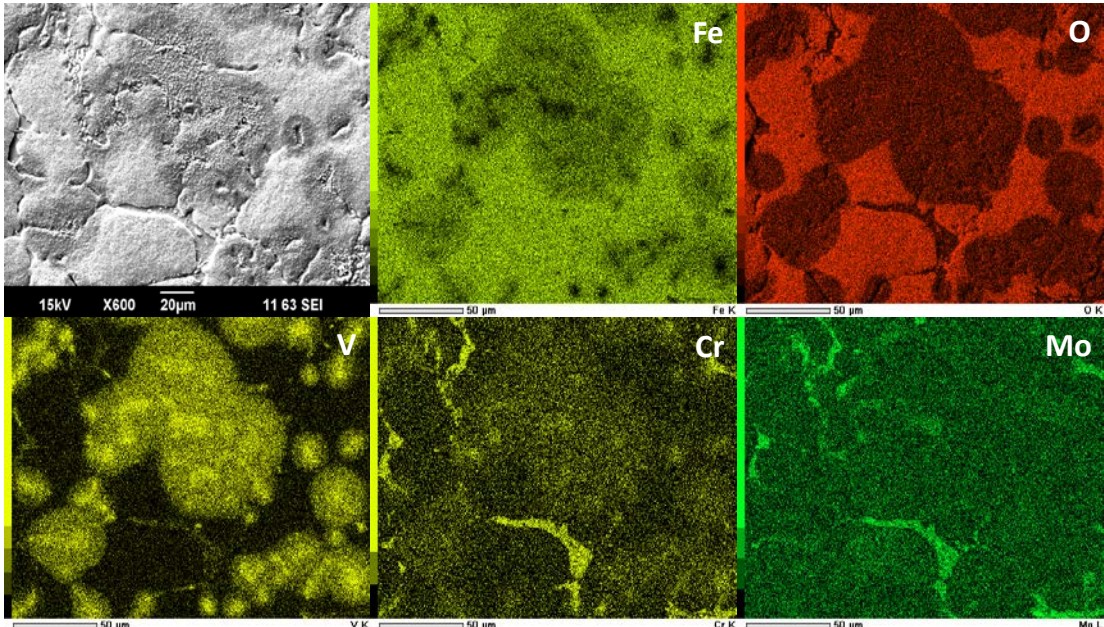

**Figure 4.** SEM X-ray maps of the HSS oxidized at 700 °C in dry air.

### 3.3.2. Oxidized in the Humid Air

Figure 5 shows SEM images of the IC roll surface morphologies oxidized in the humid air at 650 °C and 700 °C in the humid air. The IC samples' surfaces were more rough when they oxidized in the humid air than when they oxidized in the dry air. The water vapor accelerates the oxidation for both the cementite and martensite. In addition, the oxide scales formed in the humid air adhered firmly to the matrix without the presence of cracks in the oxide scales. The graphite was nearly covered by the extension of the oxide scales nearby.

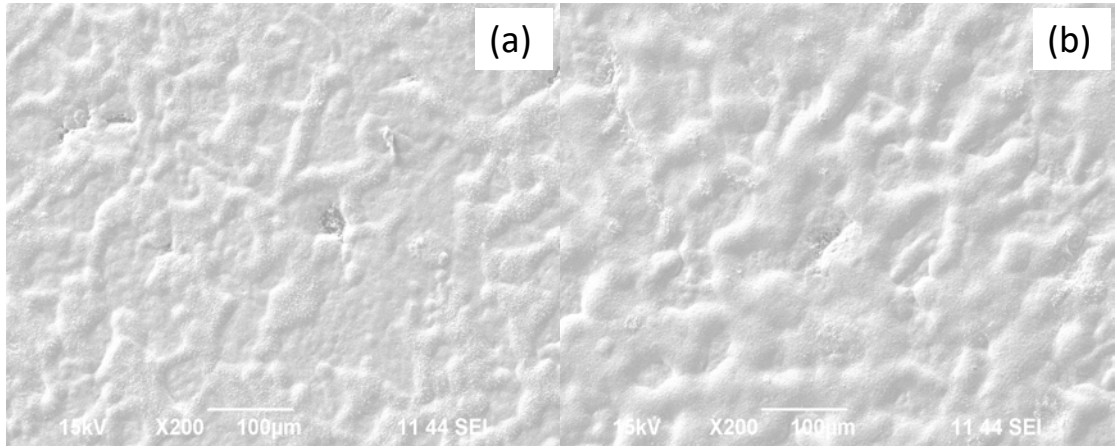

**Figure 5.** SEM images of the IC surface morphologies oxidized in humid air at (**a**) 650 °C and (**b**) 700 °C.

The HSS samples oxidized faster at 650 °C and 700 °C in the humid air. SEM X-ray maps of the HSS oxidized at 700 °C in the humid air are presented in Figure 6. The similar phenomena were observed as its counterpart in the dry air: the areas with low O Ka intensities correspond to the areas of the carbides, while the areas with high O Ka intensities are of the matrix; MC carbides are oxidized to extend the areas nearby while $M_7C_3$ carbides are still not entirely covered by the oxide scales.

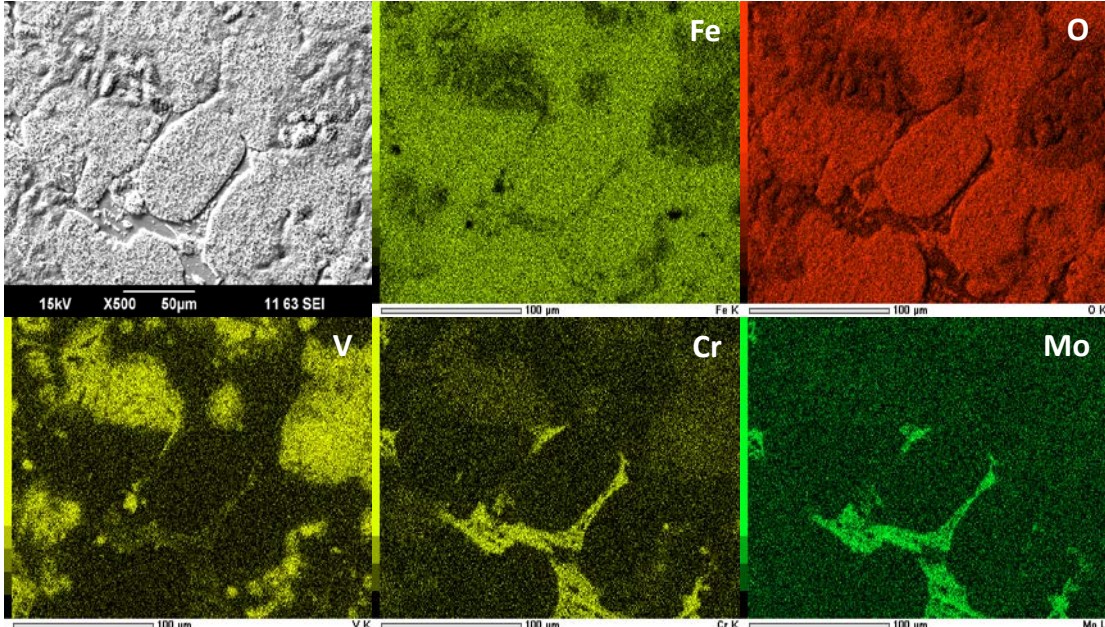

**Figure 6.** SEM X-ray maps of the HSS oxidized at 700 °C in humid air.

## 4. Cross Section Examination

The isothermal oxidation curves (Figure 2) demonstrate that the HSS oxidized faster than the IC in the dry air, whereas the IC rolls showed higher kinetics than their HSS counterparts in the humid air. In addition, the oxidation kinetics of the IC and HSS reveal a linear law and parabolic law, respectively.

Figure 7 shows SEM images of the IC cross section of the matrix regions and the graphite areas oxidized at 700 °C in both dry and humid air. As the IC rolls oxidized in the dry air (Figure 7a,b), two relatively thin oxide scales can be observed, namely a thin Cr-rich $(FeCr)_3O_4$ layer next to the substrate, and an outer layer of hematite $(Fe_2O_3)$ [31]. The oxidation of graphite is known as decarburization and left an amount of cavities, which were filled by the extension of the oxide scales. When the IC rolls oxidized in the humid air, the thickness of the oxide scales were increased from 2.26 um in the dry air to about 22.88 um in the humid air. SEM X-ray maps of the IC rolls oxidized at 700 °C in the humid air (Figure 8) reveal that both Si and Ni are present only in the substrate below the inner oxide scales, while the Cr intensifies in the inner oxide.

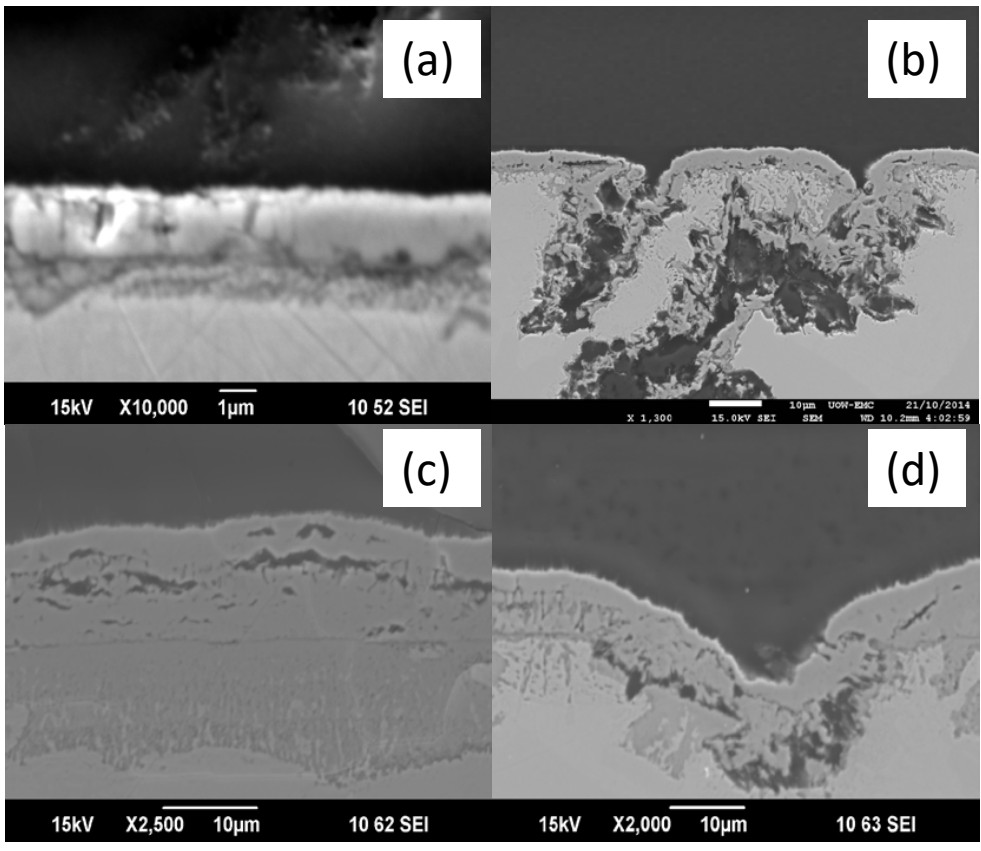

**Figure 7.** SEM images of the IC roll cross section oxidized at 700 °C: (**a**) the matrix regions and (**b**) the graphite areas in dry air, and (**c**) the matrix regions and (**d**) the graphite areas in humid air.

Figure 9 shows the cross sections and corresponding EDS line scanning analysis of the HSS oxidized at 650 °C in both dry air and humid air. The thickness of the oxide scales was insignificantly increased from 2.54 um in the dry air to 4.56 um in the humid air, which was far less obvious than the thickness increase in the IC roll oxide scales. Two oxide phases can be observed from the EDS line scanning analysis (Figure 9a)—a thin Cr-rich $(Fe,Cr)_3O_4$ next to the substrate and a thick oxide layer of $Fe_2O_3$ at the scale–gas interface, while three oxide phases, Cr-rich $(Fe,Cr)_3O_4$, magnetite and hematite, can be identified from Figure 9b. Garza et al. [22,34,35] also reported the two oxide phases formed on the HSS oxidized in the dry air, and three oxide phases in the humid air. In addition, the oxide scales tightly adhere to the matrix and no voids are found at the matrix interface.

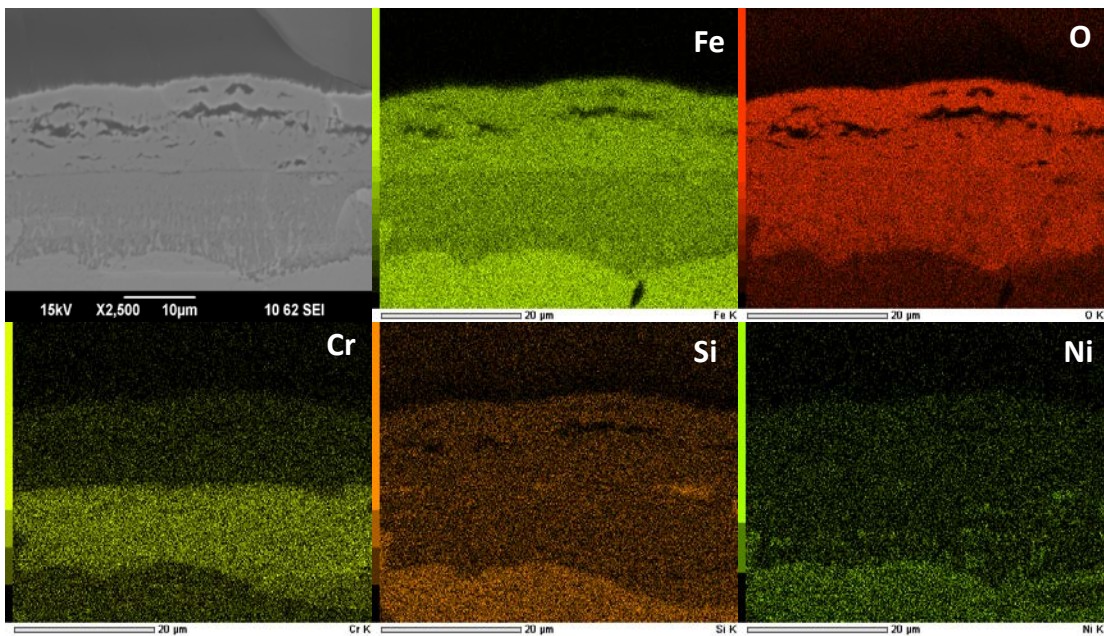

**Figure 8.** SEM X-ray maps of the IC oxidized at 700 °C in humid air.

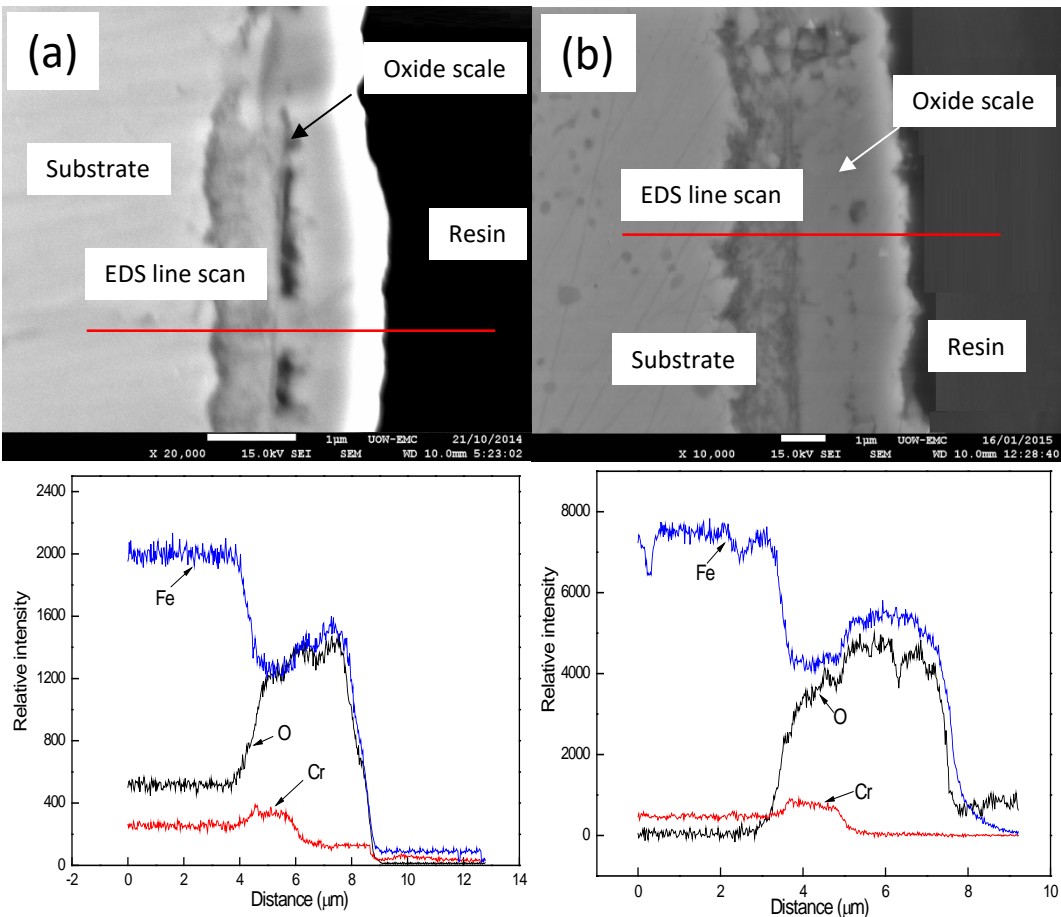

**Figure 9.** Cross sections and corresponding energy dispersive spectrometer (EDS) line scanning analysis of the HSS oxidized at 650 °C in (**a**) dry air and (**b**) humid air.

The V-rich MC carbides account for the largest fraction of all carbides and a faster oxidation was observed. Figure 10 presents SEM X-ray maps of the cross section for the MC oxidized at 700 °C in the humid air. It is evident that the vanadium oxides extend laterally, which is also confirmed by their surface morphologies in Figures 4 and 6. The possible oxidation process of vanadium-rich carbides is expressed as follows:

$$VC(s) + O_2(g) = VO(s) + CO(g) \tag{1}$$

$$2VC(s) + 3.5O_2(g) = V_2O_5(s) + 2CO(g) \tag{2}$$

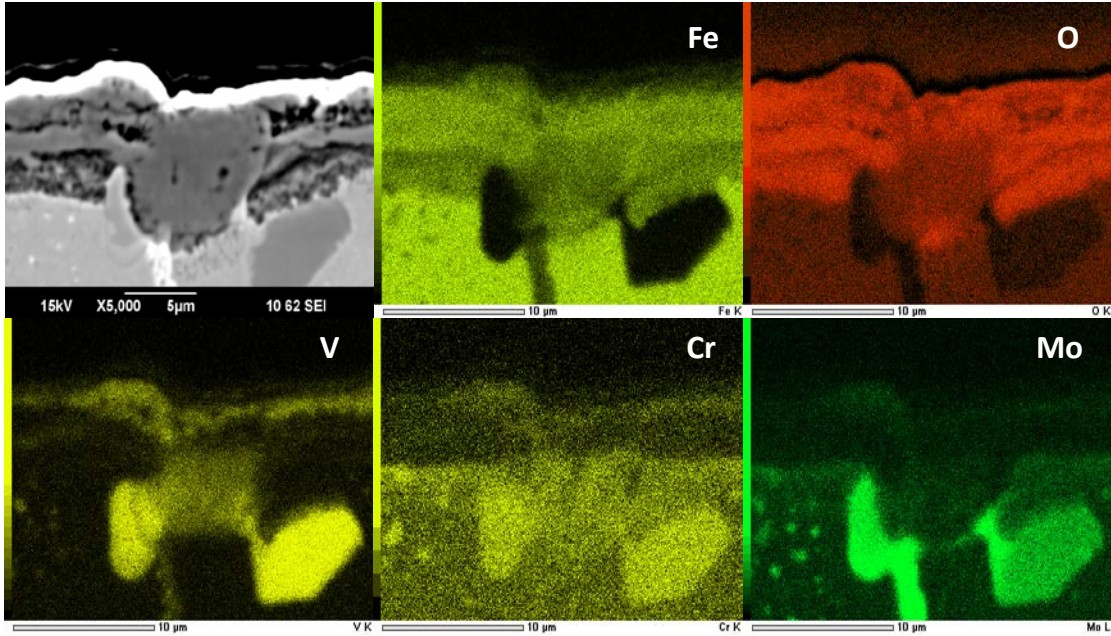

**Figure 10.** SEM X-ray maps of the cross section for the MC oxidized at 700 °C in humid air.

Some researchers [36,37] proposed that the Gibbs free energy involved in the oxidation of vanadium carbide shows that the formation of $V_2O_5$ ($\Delta G = -1357$ kJ mol$^{-1}$, 650 °C) is more favored than that of VO ($\Delta G = -457$ kJ mol$^{-1}$, 650 °C). Therefore, their conclusion was that the oxidation product of vanadium carbide is $V_2O_5$. However, the values of $\Delta G$ are expressed as kJ mol$^{-1}$ $O_2$ so that the stabilities of various oxides may be compared directly [38]. In this case, our study indicates that the formation of VO ($\Delta G = -443$ kJ mol$^{-1}$ $O_2$, 700 °C) is more preferential than that of $V_2O_5$ ($\Delta G = -383$ kJ mol$^{-1}$ $O_2$, 700 °C).

## 5. Conclusions

This isothermal oxidation investigation of the IC and HSS materials has been carried out on a thermogravimetric analyzer (TGA) at 650 °C and 700 °C for 30 min in both dry air and humid air. The following conclusions can be obtained:

1.  The HSS exhibits faster oxidation kinetics than the IC in the dry air, whereas the IC rolls oxidized more easily than their HSS counterparts in the humid air. The oxidation kinetics of the IC and HSS materials reveal a linear law and parabolic law, respectively. The oxidation of the HSS suggests that the diffusion of metal cations or oxygen anions is the rate controlling step.
2.  The oxide scales of the IC after the oxidation in the dry air and humid air are comprised of two oxide phases: an inner thin Cr-rich (Fe, Cr)$_3$O$_4$ adjacent to the matrix and an outer Fe$_2$O$_3$ at the oxide–gas interface. The oxidation of graphite was faster in the humid air than in the dry air, and denuded cavities were nearly filled by the extension of the oxide scales.

3.  Two oxide phases, $(Fe, Cr)_3O_4$, next to the matrix and an outer layer of hematite, were found when the HSS oxidized in the dry air, while three oxide phases, $(Fe, Cr)_3O_4$, magnetite and hematite, were identified in the humid air. The oxidation of V-rich MC carbides was observed in all tested cased. The Gibbs free energy for the oxidation of vanadium carbides indicates that the formation of VO is more preferential than $V_2O_5$.

**Author Contributions:** Conceptualization, G.Z.; Formal analysis, Q.D.; Investigation, Z.X.; Supervision, T.L.; Writing—original draft, L.H. All authors have read and agreed to the published version of the manuscript.

**Funding:** This research was funded by Natural Science Foundation of China (grant number: 51904217) and Natural Science Foundation of Shaanxi Province (grant number: 2020JQ-294).

**Acknowledgments:** The authors are grateful for the financial support from the National Natural Science Foundation of China (Grant No. 51904217) and Natural Science Foundation of Shaanxi Province (Grant No. 2020JQ-294).

**Conflicts of Interest:** The authors declare no conflict of interest.

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
