# Peer review of "The Oxidation Behaviors of Indefinite Chill Roll and High Speed Steel Materials"

_metals, doi:10.3390/met10081095_

Round 1

Reviewer 1 Report

The isothermal oxidation investigation of the IC and HSS materials have been carried out at 650 ℃ and 700 ℃ for 30 min in both dry air and humid air. The investigation and results are meaningfull. Before publication additional improvements are needed:

Following references should be included in Introduction and Discussion:

FABIENNE DELAUNOIS, VICTOR IOAN STANCIU and MARIO SINNAEVE, Resistance to High-Temperature Oxidation and Wear of Various Ferrous Alloys Used in Rolling Mills, METALLURGICAL AND MATERIALS TRANSACTIONS A, 49A (2018) 822-835.

David Bombač, Marius Gintalas, Goran Kugler 1 and Milan Terčelj, Mechanisms of Oxidation Degradation of Cr12 Roller Steel during Thermal Fatigue Tests, Metals 2020, 10(4), 450; https://doi.org/10.3390/met10040450.

Conclussion should be improved.

Author Response

The authors are very appreciated to the reviewer’ careful review of our manuscript and thank them for providing us with their comments and suggestions to improve the quality of the manuscript. The detailed responses have been made in the attached file.

Reviewer 2 Report

the present paper entitled 'The oxidation behaviors of indefinite chill roll and high speed steel materials' presents the oxidation description of both alloys at 650 C and 700 C in dry and humid air atmospheres.

Introduction is good with an overview of recent work in this domain.

Results shows the weight gain curves and surface characterization of both alloys in both atmospheres.

Discussion can be improved specially for the HSS alloy. Is it possible to describe the iron oxide on top of CrFe oxide layer in contact with the matrix.

Do you see voids at the matrix interface?

Overall, a nice work and smooth paper to read.

Author Response

(The authors gave the same response as above.)

Reviewer 3 Report

Reviewers comments.

The paper presents oxidation data on two alloys used in the sheet rolling industry.  The components experience rapid heat exposures for short periods of time.  A limited study has been performed with four samples exposed to two temperatures of relevance and two environments.  The results demonstrate that the alloy with higher Cr and V content oxidizes at a faster rate than the lower Cr, no V alloy in dry air.  The Cr and V containing alloy shows no difference in oxidation rate between the two environments but the low Cr alloy showed a marked increase.

The paper would benefit from more data points, i.e. testing of more samples.  This would give greater confidence the results presented and demonstrate variability. 

More detailed comments with corrections are given below.  English should be improved to prevent misunderstanding.

Abstract

“ The results indicated that the HSS was oxidized more severely than the IC in the dry air, whereas the opposite occurred in humid air.  the IC 11 was more easily to be oxidized than the HSS in the humid air. “ In addition, the results do not show “severe” oxidation in any of the samples. Faster kinetics could be claimed.  Adjust the working appropriately and consistently throughout the paper.

Introduction

 However, the absence of no carbides in the low-carbon rolls led to low wear resistance with and a frequent replacement of roller required.

 They This improved the present high wear resistance and strength, but still difficult reduced the oxidation rates to be oxidized.

The introduction is well constructed and includes relevant papers.

Experimental

The authors refer to another paper to get details of the experimental detail but more is needed here.  For instance: is a buoyance correction applied to this data, has analysis of the data been performed to determine if the data is linear or parabolic, does the data show parabolic rates across the entire oxidation time?

How do they introduce the moisture and how true to work conditions is it?

57  “The IC roll mad  and HSS materials.”

63 “with a vertical hang-down SETSYS balance. The oxidation tests were conducted under two

64 atmospheres, 20% water vapor and dry airs air.”

70 “kinematic kinetics curves were obtained by plotting mass gain change per surface against the oxidation time.”

3.2. Oxidation kinematic  kinetics

The isothermal oxidation curves of the IC and HSS materials in both dry and humid airs air are shown in Fig. 2. It is evident that both the temperature and oxidizing atmospheres obviously

 85 “influence the mass gain of the materials. Overall, the curves go up with the increase of the oxidation 86 temperature.”

3.3.1. Oxidization in the dry air

I agree with the observations of the surface of the samples presented in Figure 3.  However, the information provided by EDS maps, as presented in Figure 4 for the HSS samples needs much care in interpretation.  Some of the compositional differences come from the underlying alloy due to the interaction volume of beam with the sample and the thickness of the surface oxides.  The distortion in the compositional data is made more difficult to interpret with oxides of different thicknesses across the surface.

“Similar phenomena were observed for the HSS oxidized at 650oC and 700oC in the dry air. “SEM X-ray maps of the HSS oxidized at 700 oC in the dry air are presented in Fig. 4, from which it is seen

107 that the areas of low O Ka intensity correspond to the areas of the carbides while the areas with the 108 high O Ka intensity are of the matrix.”  -  This could be due to a difference oxide composition with different thicknesses.  This is better demonstrated in cross section.

“The oxides of MC carbides” – I think you mean the oxide formed above the MC carbides.

extend to the around areas, laterally.

 whereas 109 the M7C3 carbides are still not totally covered by the oxides. – This might not be true, it could be that there is a very thin oxide on the surface, especially if these are Cr-rich carbides as you state in the introduction.

“Therefore, the matrix is oxidised more 110 easily readily than that of the carbides, and the M7C3 carbides are resistance to oxidation harder to be oxidised than the MC carbides.” – this is not necessarily the case.  It could just be the difference in the kinetics that you are detecting. 

3.3.2. Oxidation in the humid air

The authors state that the HSS alloy oxidizes more severely in humid air but there is not real evidence for this in these micrographs.  The word “severely” needs to tempered to “faster”.

The authors move to a discussion section where they present more of their results.  This should be re-titled as, e.g., Cross-sectional examination. 

  1. Discussion

“The isothermal oxidation curves (Fig. 2) demonstrated that the HSS was oxidized more severely faster

133 than the IC in the dry air, whereas the IC showed higher kinetics was more easily to be oxidized than the HSS counterparts

134 in the humid air. In addition, the oxidation kinetics of the IC and HSS reveal a linear law and parabolic 135 law, respectively.”

“When the IC was oxidized in the humid air, the thickness of the oxide scales were increased from 2.26 m in the dry air to about 22.88 m in the humid air. SEM X-ray maps of the IC oxidized at 700 oC 143 in the humid air (Fig. 8) reveal that both Si and Ni only present in the substrate below the inner oxide 144 scales, while the Cr intensifies in the inner oxide.” – the authors need to examine the EDS maps more fully as there is Fe present too.  The concentrations obtained from EDS maps can be distorted by changing the contrast and brightness.  It would be better to include true compositional data to determine the actual concentration of the alloy.  This will aid in the interpretation of the oxidation behavior of the two alloys.  Can the authors confirm whether magnetite is also present, the EDS maps suggest it is.

The use of linescans to indicate the change in composition of the oxides present is much more convincing than the maps used for the IC alloy.  Compositional data, in wt.% or at.% would be better and as at.% the data could help in identifying whether it is Fe3O4 or Fe2O3.  The authors need to provide the line scan traces for both sets of samples with compositions for the alloy and oxides.  In addition, depletion profile of the alloy from the interface from the oxide into the bulk will show the impact of the oxidation on the depth of compositional change.

“Since the V-rich MC carbides account for the largest fraction of all carbides and severe oxidation 159 was observed.”  This is not a sentence change.

Figure 10, include alloy in captions to aid the reader.  Do this for all captions.

“Figure 10 presents SEM X-ray maps of the cross section for the MC oxidized at 700 oC 160 in the humid air. It is evident that the vanadium oxides extend to their nearby laterally, which is also 161 confirmed by their surface morphologies in Figures 4 and 6.”   This confirmation of the description of Figures 4 and 6 earlier is good and thus I am happy for the earlier comments on the EDS maps but the authors must add in the fact that the maps are collecting information from the underlying alloy as well as the oxide.

Author Response

(The authors gave the same response as above.)
